# The Complement Component 4 Binding Protein α Gene: A Versatile Immune Gene That Influences Lipid Metabolism in Bovine Mammary Epithelial Cell Lines

**DOI:** 10.3390/ijms25042375

**Published:** 2024-02-17

**Authors:** Xuanxu Chen, Zhihui Zhao, Xinyi Jiang, Jing Li, Fengshuai Miao, Haibin Yu, Ziwei Lin, Ping Jiang

**Affiliations:** 1The Key Laboratory of Animal Genetic Resource and Breeding Innovation, College of Coastal Agricultural Sciences, Guangdong Ocean University, Zhanjiang 524088, China; chenxuanxugdou@163.com (X.C.); zhzhao@gdou.edu.cn (Z.Z.); jxy991130@163.com (X.J.); lijing0305@outlook.com (J.L.); miaofengshuai@163.com (F.M.); yuhb@gdou.edu.cn (H.Y.); 2The Key Laboratory of Animal Resources and Breed Innovation in Western Guangdong Province, Zhanjiang 524088, China

**Keywords:** BMECs, *C4BPA*, CRISPR/Cas9, lipid metabolism

## Abstract

Complement component 4 binding protein α (*C4BPA*) is an immune gene which is responsible for the complement regulation function of *C4BP* by binding and inactivating the Complement component C4b (*C4b*) component of the classical Complement 3 (*C3*) invertase pathway. Our previous findings revealed that *C4BPA* was differentially expressed by comparing the transcriptome in high-fat and low-fat bovine mammary epithelial cell lines (BMECs) from Chinese Holstein dairy cows. In this study, a *C4BPA* gene knockout BMECs line model was constructed via using a CRISPR/Cas9 system to investigate the function of *C4BPA* in lipid metabolism. The results showed that levels of triglyceride (TG) were increased, while levels of cholesterol (CHOL) and free fatty acid (FFA) were decreased (*p* < 0.05) after knocking out *C4BPA* in BMECs. Additionally, most kinds of fatty acids were found to be mainly enriched in the pathway of the biosynthesis of unsaturated fatty acids, linoleic acid metabolism, fatty acid biosynthesis, and regulation of lipolysis in adipocyte. Meanwhile, the RNA-seq showed that most of the differentially expressed genes (DEGs) are related to PI3K-Akt signaling pathway. The expressions of 3-Hydroxy-3-Methylglutaryl-CoA Synthase 1 (HMGCS1), Carnitine Palmitoyltransferase 1A (CPT1A), Fatty Acid Desaturase 1 (FADS1), and Stearoyl-Coenzyme A desaturase 1 (SCD1) significantly changed when the *C4BPA* gene was knocked out. Collectively, *C4BPA* gene, which is an immune gene, played an important role in lipid metabolism in BMECs. These findings provide a new avenue for animal breeders: this gene, with multiple functions, should be reasonably utilized.

## 1. Introduction

Dairy farming holds a pivotal role in modern animal husbandry and the food industry. Milk and dairy products are crucial to the human diet, providing protein, fatty acids, calcium, potassium, and so on [1]. Milk fat trait is the main factor affecting milk quality, and it is also the target trait in the molecular breeding of dairy cows. Bovine mastitis, characterized by the inflammation of the mammary glands, is one of the most significant diseases affecting dairy farming. It has a negative effect on both animal welfare and food security [2]. Mammary epithelial cells (MECs) play an important role in the process of milk biosynthesis. Due to the presence of a large number of other types of cells around MECs (such as adipocytes, macrophages, neutrophils), it is very common to use breast tissue for omics analysis and obtain functional genes related to milk fat and immune regulation at the same time. Notably, in our previous study, in the absence of interference from other types of cells, the transcriptome analysis of high-fat and low-fat bovine mammary epithelial cell lines revealed that the complement component 4 binding protein α (*C4BPA*) (related to immune regulation) were differentially expressed [3]. During animal breeding, it is very common for a gene to regulate multiple traits. Hence, researchers and breeders are very concerned about this type of gene with multiple functions.

The complement system is an important component of defense against foreign organisms, involving both the innate and adaptive immune systems [4]. Complement component 4 binding protein (*C4BP*) is a regulator of the classical complement and lectin complement pathways [5]. *C4BP* has different combinations of α and β chains. Thus, it exists in three different subtypes: *α_7_β_1_*, *α_7_β_0_*, and *α_6_β_1_* [4]. During inflammation, it has been observed that the plasma levels of C4BPα chains increased, but not those of C4BPβ chains [6,7]. Additionally, Complement component 4 binding protein α (C4BPα) chains also contain binding sites of Cluster of differentiation 40 (CD40), complement component c3b (C3b), low density lipoprotein receptor associated protein (LRP), and some bacterial surface proteins, which are key molecules involved in the inflammation, lipid metabolism, and coagulation pathways [4,8]. The *C4BPA* gene, situated on chromosome 1q32, consists of 12 exons and 11 introns [9,10]. A study reported that *C4BPA* may be produced by the host immune response against tumors in the sera of pancreatic ductal adenocarcinoma patients [11]. Moreover, the relative content of ten proteins, including Component 2 (C2), C3, Component 5 (C5), and *C4BPA*, in the serum of dogs with obesity syndrome increased [12]. And our previous study found that the *C4BPA* gene had a significant difference in the mRNA expression of high fat and low fat cows [13]. Our other study demonstrated that the *C4BPA* gene is a co-regulator of immunity and fat metabolism in BMECs via transfecting the *C4BPA* knockdown vector and the overexpression vector [10].

Therefore, the purpose of the present study was to investigate the effects of the *C4BPA* gene on lipid metabolism in bovine mammary epithelial cells (BMECs), using the CRISPR/Cas9 technique to establish a *C4BPA* knockout BMECs line model. This study will be valuable for applying the *C4BPA* gene, which is an immune related gene, in breeding for increased milk quality; additionally, the findings may contribute to the knowledge of the comprehensive selection of relevant traits in the breeding of milk fat traits and immune traits in cows.

## 2. Results

### 2.1. Construction of C4BPA Knockout BMECs Line Model

Utilizing CRISPR/Cas9 technology, we designed sgRNAs (Figure 1A) and transfected the resulting recombinant plasmid into BMECs (Figure 1B). Upon the use of agarose gel, the WT-BMECs group and the KO-BMECs group demonstrated a 656 bp band and a 296 bp band, respectively, for the *C4BPA* PCR products (Figure 1C). Lines 1–9 represent WT-BMECs which are the knockout-failed sequence (negative control) and line 10 corresponds to KO-BMECs. Additionally, genomic DNA isolated from both cell types underwent sequencing, confirming the effective disruption of *C4BPA* target sequences in BMECs (Figure 1D). Compared to the WT-BMECs, the mRNA and protein expression of *C4BPA* significantly reduced in KO-BMECs via using RT-qPCR and ELISA separately (*p* < 0.05) (Figure 1E,F). Furthermore, amino acid sequence prediction based on DNA sequences revealed distinctions between the protein sequences of WT-BMECs and KO-BMECs (Figure 1G).

### 2.2. The Proliferation Rate and Apoptosis Rate of KO-BMECs and WT Cell Line

Apoptosis represents a principal mechanism governing programmed cell death and proliferation. As shown in Figure 2A,B, knocking out of *C4BPA* gene exhibited no impact (*p* > 0.05) on BMECs apoptosis. Concurrently, the findings from the CCK-8 assay indicated that the proliferation rate of KO-BMECs did not demonstrate a statistically significant increase compared to that of WT-BMECs over 0–48 h (Figure 2C). These results indicated that this cell line could be used for further studies.

### 2.3. Lipid Accumulation in BMECs of C4BPA Gene KO or WT Cell Line

Lipid accumulation in KO-BMECs and WT-BMECs were analyzed. As shown in Figure 3, the average fluorescence intensity of KO-BMECs was significantly higher than that in the WT-BMECs group (*p* < 0.05). The results demonstrated that knocking out of *C4BPA* gene enhanced the accumulation of cellular lipid droplets.

### 2.4. Triacylglycerol (TG), Cholesterol (CHOL), and Free Fatty Acid (FFA) Contents in BMECs of C4BPA Gene KO or WT Cell Line

TG, CHOL, and FFA content in KO-BMECs and WT-BMECs was compared. The results showed that the TG content in KO-BMECs increased compared with that in WT-BMECs (*p* < 0.05) (Figure 4A). However, the CHOL and FFA content of KO-BMECs was lower than that of WT-BMECs (Figure 4B,C).

### 2.5. Results of Fatty Acid Contents in BMECs of C4BPA Gene KO or WT Cell Line Based on the GC-MS

The studies described above found that the FFA contents decreased after knocking out of *C4BPA* gene. Therefore, GC-MS experiments were used to further explore the effect of *C4BPA* on fatty acid metabolism. In saturated fatty acids, after *C4BPA* gene deletion, the expressions of caprylic acid, heptadecanoic acid, palmitic acid, arachidic acid, and stearic acid remarkably increased; meanwhile, myristate acid and behenic acid decreased significantly (Figure 5A,B). In unsaturated fatty acids, the expression levels of Cis-11-octadecenoic acid, palmitelaidic acid, and all-5,8,11,14-eicosatetraenoic acid increased significantly, while linolenic acid, palmitoleic acid, and linoleic acid decreased significantly. KEGG analysis showed that, after knocking out *C4BPA*, most kinds of the fatty acids are mainly enriched in the linoleic acid metabolism, fatty acid biosynthesis, and bio-synthesis of unsaturated fatty acids. These results suggested that *C4BPA* gene knockout may affect the biological processes related to lipid metabolism (Figure 5C,D).

### 2.6. Identification of DEGs in BMECs of C4BPA Gene KO or WT Cell Line

RNA-seq analysis was used to explore the function of *C4BPA* gene. A total of 1326 differentially expressed genes (DEGs) were found, including 672 upregulated genes and 654 downregulated genes (Figure 6A–C). To verify the transcriptome results, 10 genes were randomly selected for qPCR verification. The results showed that knocking out of *C4BPA* upregulated three genes: Serine Protease 23 (*PRSS23*), Lipoprotein Lipase (*LPL*), and Ornithine Decarboxylase 1 (*ODC1*) (Figure 6D). In addition, knocking out of the *C4BPA* gene also caused the downregulation of five genes: Fatty Acid Desaturase 1 (FADS1), Interleukin 6 (*IL-6*), Interferon Alpha Inducible Protein 27 (*IFI27*), Interleukin 1 Beta (*IL1B*), and Carnitine Palmitoyl transferase 1A (*CPT1A*). Meanwhile, two genes showed no significant changes, namely Acyl-CoA Synthetase Long chain Family Member 1 (*ACSL1*) and Acyl-CoA Dehydrogenase Short Chain (*ACADS*) (Figure 6D). These results were also consistent with the RNA-seq results, demonstrating the RNA-seq’s reliability.

### 2.7. Functional Enrichment Analysis of the DEGs in BMECs of C4BPA Gene KO or WT Cell Line

As shown in Figure 7A, the KEGG results showed that most of the DEGs are associated with the PI3K-Akt signaling pathway and pathways in cancer. And the GO analysis showed that the molecular function focused on calcium ion binding (Figure 7B).

### 2.8. Expression of Different Genes Related to Fat Metabolism

As shown in Figure 8, compared to the WT-BMECs, the expression of HMGCS1, CPT1A, and FADS1 significantly decreased in the KO-BMECs group (*p* < 0.05), while the expression of SCD1 increased (*p* < 0.05).

## 3. Discussion

With the continuous improvement of living standards, people’s demand for good quality milk has been rapidly increasing. Mastitis, a common disease in cows, leads to reduced production, decreased quality, and increased veterinary costs [14]. Mastitis poses a huge threat to animal and human health, and causes significant economic losses to modern dairy farming and dairy industry [15]. Therefore, it is crucial to cultivate cows that are resistant to mastitis. Milk fat is one of the main substances that comprises the nutrition of milk. TG, CHOL, FFA, and some other fat soluble vitamins are the important components of milk fat. Sick cows will exhibit significant abnormal changes in both their breasts and milk. When lactating cows develop mastitis, the milk fat content in will decrease. There is a close relationship between milk quality traits and breast inflammation and the genes related to immune regulation play pivotal roles in fat metabolism regulation. Therefore, dairy breeders focus on selecting genes that can increase milk fat while reducing susceptibility to mastitis. This study aimed to explore the role of *C4BPA* gene which as an immune gene in lipid metabolism using *C4BPA* knockout BMECs line model.

CRISPR/Cas9, a gene editing technique, provides efficient, easy, and fast identification and editing of the target gene [16]. Currently, CRISPR/Cas9 technology can knockout target genes at the genetic level effectively, so it can induce gene knockout in the cell [16]. Recent research has utilized CRISPR/Cas9 technology to knockout or knockdown various functional genes, aiding researchers in comprehending their cellular level functions across diverse metabolisms, pathways, and biological processes. Jin et al. [17] used the CRISPR/Cas9 system to establish a *HMGA2* knockout PTC cell line in order to exploring the *HMGA2* effect on PTC cell proliferation and invasion. A Cluster of differentiation 16 α (*CD16A*) knockout human embryonic stem (hESC) cell line was constructed and was used to investigate the role of *CD16A* in the human cell development [18]. In our study, CRISPR/Cas9 technologies were used to construct a bovine mammary epithelial cell line with knockout of *C4BPA*, which provided an effective tool for studying the function of *C4BPA*.

TG as a molecular indicator of dairy product quality and it also plays a significant role in energy storage and supply. Wu et al. [19] reported that lipopolysaccharide (LPS) could significantly reduce the content of TG and milk fat synthesis compared with the CON group in BMECs. Interestingly, *C4BPA* knockout increased the TG content in this study. Therefore, we speculated that *C4BPA* could regulate TG levels when mastitis occurs. CHOL is an essential component in animal tissue cells. It not only participates in cell membrane formation but also serves as a precursor for the synthesis of bile acids, vitamin D, and steroid hormones [20]. However, excessive intake of cholesterol in the human body may lead to cardiovascular and other diseases. Therefore, it is necessary to produce low fat and low cholesterol milk in cow breeding. In this study, knocking out of *C4BPA* could reduce the CHOL content. FFA is one of the products of fat metabolism and an important energy substance in the body’s energy metabolism. Lipid droplets, the intracellular sites for neutral lipid storage, are closely related to cellular lipid metabolism. And they have also become important nodes in the transport of fatty acids inside the cell and between cells [21]. Knocking out of the *C4BPA* gene could regulate the FFA content and the accumulation of lipid droplets. Furthermore, after knocking out the *C4BPA* gene, the altered fatty acids are enriched in the lipid metabolic pathway. The above results demonstrate that the *C4BPA* gene may influence lipid metabolism in BMECs.

In addition, the RNA-seq analysis showed that, in knocking out the *C4BPA* gene, most of the genes are enriched in the PI3K/Akt signaling pathway. The PIK3 signaling pathway represents a group of enzymes that lead to the production of lipid second messengers; these are involved in various signal transductions, including cell proliferation, apoptosis, and inflammatory cascade [22]. This process is mediated by the serine or threonine phosphorylation of a series of downstream substrates, involving key genes such as Phosphatidylinositide 3-kinases (*PI3K)* and Protein Kinase B (*AKT)*, so this pathway is directly named after these two genes. Growing evidence also indicates that the PI3K/AKT pathway is an important cellular signaling cascade for cellular defense against inflammatory stimuli [22,23]. The PI3K-Akt axis can alter the inflammatory response by recruiting and activating innate immune cells, such as macrophages [24,25]. In macrophages, the PI3K/Akt pathway transduces signals from various receptors, including cytokine and adipokine receptors, as well as receptors necessary for inducing innate immune activation. Therefore, the activation of the PI3K/Akt pathway coordinates the response of macrophages to different metabolic and inflammatory signals. Meanwhile, PI3K/AKT pathway also plays a pivotal role in glucose, lipid, and protein metabolism [26]. The adipogenic effect of insulin is mediated by the PI3K/AKT signal. After activation in mature adipocytes, the PI3K/AKT signaling pathway would affect lipid biogenesis and lipid degradation [27]. Also, the PI3K/AKT signaling pathway plays an important role in adipogenesis [28]. These findings indicate that the *C4BPA* gene may regulate lipid metabolism via the PI3K-AKT signaling pathway in BMECs.

Following the knockout of the *C4BPA* gene, five genes, including *HMGCS1*, *CPT1A*, *FADS1*, and *SCD1*, were examined. *HMGCS1*, a member of the *HMGCS* family, is involved in the different biological process of cholesterol [29]. In this study, knocking out of the *C4BPA* gene decreased the content of cholesterol and the expression of HMGCS1. Additionally, *CPT1A*, a rate-limiting fatty acid oxidation enzyme, is closely related to fat deposition by affecting lipid metabolism [30]. And the knockdown of *CPT1A* significantly upregulated the expression of adipogenic genes; meanwhile, the overexpression of *CPT1A* markedly decreased triglyceride content and lipid accumulation in goat adipocytes [30]. A study showed that overexpression of CPT1A mitigated lipid accumulation and cholesterol uptake in clear-cell renal carcinoma (ccRCC) [31]. Similarly, another study reported that moderately increasing the expression of CPT1A is sufficient to reduce hepatic TG levels [32]. In this study, knocking out *C4BPA* reduced the expression of CPT1A and increased the levels of TG; this finding it similar to those of the above studies. *FADS1* has been linked to hyperactivity, cholesterol, and triglyceride level [33]. *SCD1*, a key enzyme that catalyzes the synthesis of monounsaturated fatty acids, plays a role in several signaling pathways that are related to lipid metabolism [34]. Li et al. [35] reported that the *SCD1* gene mediated the expression of genes related to FA and TAG biosynthesis in BMECs. Our results demonstrated that the expression of SCD1 increased, and that TAG and FA contents also changed in KO-BMECs. Moreover, GC-MS showed that most kinds of fatty acids are enriched in the biosynthesis of unsaturated fatty acids, linoleic acid metabolism, and fatty acid biosynthesis after knocking out the *C4BPA* gene. The above results demonstrate that the *C4BPA* gene could regulate gene-associated lipid metabolism.

In conclusion, our findings reveal that TG, CHOL, and FFA contents significantly changed after knocking out the *C4BPA* gene. And most kinds of fatty acids were enriched in the lipid metabolism. The findings also showed that *C4BPA* is also a crucial gene in regulating genes associated with lipid metabolism, such as *HMGCS1*, *CPT1A*, *FADS1*, and *SCD1*. Moreover, the RNA-seq analysis indicated that the *C4BPA* gene may regulate lipid metabolism via the PI3K/Akt signaling pathway, laying the foundations for further study of the molecular mechanisms by which *C4BPA,* as an immune gene, affects lipid metabolism in BMECs. Hence, we speculate that the *C4BPA* gene could always be involved in lipid metabolism in BMECs. In addition, the *C4BPA* knockout BMECs line was constructed, providing a favorable tool for further study of the function of *C4BPA*.

## 4. Materials and Methods

### 4.1. Cell Culture and the Construction of C4BPA Knockout BMECs Line Model

Mammary epithelial cells were obtained from the College of Coastal Agricultural Sciences of Guangdong Ocean University. BMECs were cultured in DMEM/F12 medium (GIBCO, Grand Island, NY, USA) with 10%FBS (PAA, Parching, Austria) in a 5% CO_2_ incubator at 37 °C. The cells were cultured in 6 well plates (Falcon, Franklin Lakes, NJ, USA). Upon reaching 70–80% confluency, 3 μg of recombinant plasmid DNA, 6 μL of transfection reagent, and 200 μL of Opti MEM (GIBCO, Grand Island, NY, USA) were mixed for 15 min and transfected into the BMECs. After 24 h, the cells were collected using trypsin without EDTA solution, and were screened using flow cytometry to see whether they could express a green fluorescent protein. The single positive cells (which can express a green fluorescent protein) were cultured in 96 well plates containing 20% serum and 1% antibiotics. Seven days later, under a microscope, a small pile of cell pellets could be seen. After digesting the cells and continuing culturing, a large population of cells could be observed under the microscope 15 days later. Then, these cell clusters were identified by PCR. Although the transfection efficiency was evaluated by flow cytometry, our primary objective for cell transfection was to screen for positive cells that were successfully expressed. All experiments were carried out following Guangdong Ocean University’s guidelines for the care and use of experimental animals. WT-BMECs represented a wild-type BMECs line and KO-BMECs represented *C4BPA* knockout BMECs line.

### 4.2. Construction of C4BPA Gene BMECs KO Cell Line

The identification of single-guide RNAs (sgRNAs) targeting the *C4BPA* gene exon 4 in cattle was conducted using the online analysis tool available at https://www.benchling.com/crispr (accessed on 18 August 2021). The sgRNAs (Table 1) were synthesized by Shanghai Sheng Gong Company in Shanghai, China, ensuring specificity and proper location. The resulting double-stranded sgRNA was synthesized through annealing and subsequently inserted into the 48138-EGFP-puro plasmid, yielding the construct 43138-sgRNA. Following transfection, flow cytometry (FCM) was employed to discern green fluorescent protein positive cells, utilizing FACS Aria III (Becton, Dickinson and Company in Franklin Lakes, NJ, USA). Post culture, a micro DNA extraction kit was utilized to extract cellular DNA genomes, enabling the identification of the total cell DNA.

### 4.3. Analysis of mRNA and Protein Levels in BMECs of C4BPA Gene KO or WT Cell Line

Total cellular RNA was extracted from the cells using Trizol reagent (Tiangen Company in Beijing, China) and converted to cDNA by the Superscript First Strand Synthesis Kit (Invitrogen, Waltham, MA, USA). Real-time PCR assays were conducted with SYBR Green Real-Time PCR Master Mix (Tiangen Company in Beijing, China), employing specific primers for *C4BPA*: forward primer 5’- TCAATGGTTGTCTTGGCTTAC-3’ and reverse primer 5’-CTGGGCAACATACCTCCTT-3’. The PCR primers, designed based on the gene sequence, were as follows: C4-PCR-F: 5’-CCAAGTACACACTGGAGATGTAGAATGGAGCT-3’, C4-PCR-R: 5’-CCTGTGCAGGCGTAAAAGGGAATTAATGAG-3’. The total length of the amplified fragment by the identified primer was 656 bp and the length of the sequence between the two designed sgRNAs was 360 bp. If the target fragment was disrupted, then a band of 296 bp was observed in the electrophoresis result; conversely, if the target fragment remained intact, then a band of 656 bp was detected. A portion of genomic DNA extracted from the amplified cells was used for the PCR verification of knockout results. The *C4BPA* mRNA level was determined by normalizing the threshold cycle of the *C4BPA* gene to the negative control using the formula (2^−ΔΔCT^ as the relative expression ratio). The supernatant was extracted, and the total protein content of the cells was assessed using a bovine *C4b* binding protein α chain ELISA kit (Vazyme Biotech Company in Nanjing, China) along with enzyme markers. Additionally, genomic DNA isolated from WT-BMECs and KO-BMECs underwent sequencing for further confirmation.

### 4.4. Cell Apoptosis in BMECs of C4BPA Gene KO or WT Cell Line

The cell cultures were established in 6 well plates. After a 36 h incubation period, the cells were treated with the Annexin V-FITC apoptosis detection kit (Vazyme Biotech Company in Nanjing, China), following the manufacturer’s instructions. The percentage of apoptotic cells, specifically stem cells (SCs), was determined using flow cytometry. Subsequently, cells were dissociated using 0.25% Edta free trypsin, and digestion was halted by the addition of complete medium once cell detachment commenced. The cells were then suspended in 100 μL binding buffer and transferred to flow tubes. To each tube, 5 μL of PE dye and Annexin V-FITC dye were added, thoroughly mixed, and incubated for 10 min in the dark. Following light shielding, 400 μL of binding buffer was introduced to each flow tube, and apoptosis was assessed by the machine within 1 h.

### 4.5. Lipid Droplet Accumulation in BMECs of C4BPA Gene KO or WT Cell Line

KO-BMECs and WT-BMECs were cultured in 6 well plates and treated with a BODIPY working solution for 30 min to facilitate dark staining. Subsequently, the cells were fixed with 4% paraformaldehyde. Following fixation, DAPI was applied to stain the nuclei, and the cells were incubated at 37 °C for 10 min. The staining outcomes were assessed using a confocal microscope after a thorough PBS cleaning process. The fluorescence intensity of KO-BMECs was then compared with that of WT-BMECs.

### 4.6. Triglycerides, Cholesterol, and FFA Content in BMECs of C4BPA Gene KO or WT Cell Line

Cell lysis was achieved through ultrasound, and the total protein extraction followed the previously outlined procedure. The quantification of triglycerides, cholesterol, and free fatty acids (FFA) in KO-BMECs and WT-BMECs was conducted using the Triglyceride Kit, the Cholesterol Kit, and the FFA Kit (Applygen Technologies Inc. in Beijing, China). Optical density (OD) values were determined using a microplate reader (SM600, Shanghai, China). Simultaneously, the concentration of total protein was measured using the NanoDrop 2000 (Thermo Fisher Scientific Company in Waltham, MA, USA).

### 4.7. Intracellular Fatty Acids Content in BMECs of C4BPA Gene KO or WT Cell Line

After 12 h of cell culture, the spent medium was aspirated from the Petri dish. The cells were washed three times with PBS solution to eliminate dead cells. The subsequent step involved trypsin digestion and terminated digestion, with the resultant cell precipitates collected in a 50 mL centrifuge tube. A measure of 150 μL of water was added to the sample. After swirling for 30 s, the sample was frozen with liquid nitrogen and then thawed three times. The samples were sonicated in an ice water bath for 10 min. A measure of 50 μL of homogenate was used to determine the protein concentration. A measure of 400 μL extracting solution was added (V_Isopropanol_:V_n-Hexane_ = 2:3, containing 0.2 mg/L internal standard), then the mixture was vortex mixed for 30 s. Ultrasound treatment was conducted for 10 min (incubation with ice water). Then, it was centrifuged with 12,000 rpm at 4 °C for 15 min. The supernatant was transferred into 2 mL EP tubes. A measure of 500 μL extracting solution (V_Isopropanol_:V_n-Hexane_ = 2:3, containing 0.2 mg/L internal standard) was added and the mixture was vortex mixed for 30 s. Ultrasound treatment was conducted for 10 min (incubation with ice water). The mixture was centrifuged at 12,000 rpm at 4 °C for 15 min. The supernatant was mixed with the supernatant from the previous step, and the combination was vortexed for 10 s. Then, the supernatant was transferred into 2 mL EP tubes. It was then blown dry using nitrogen. A measure of 500 μL of methanol/trimethylsilyl diazomethane solution (1:2) was added; then, the mixture stood at room temperature for 30 min. It was then blown dry using nitrogen. A measure of 160 μL of n-Hexane was added; then, it was redissolved and centrifuged at 12,000 rpm for 5 min. The supernatant was transferred to a fresh vial for GC-MS analysis.

### 4.8. GC-MS Analysis in BMECs of C4BPA Gene KO or WT Cell Line

An Agilent 7890 B gas chromatograph system which utilized a DB-FastFAME capillary column, coupled with the Agilent 5977 B mass spectrometer, was used for GC-MS analysis. The sample was injected in split mode (5:1) with 1 μL aliquot. Helium gas was used as the carrier gas, with a front inlet purge flow rate of 3 mL min^−1^. The flow rate through the chromatography column at constant pressure was 46 psi. The initial temperature was maintained at 50 °C for 1 min; this was raised to 200 °C at 50 °C min^−1^ for 15 min; then, it was raised to 210 °C at 2 °C min^−1^ for 1 min; then, it was raised to 230 °C at 10 °C min^−1^ for 15 min. The temperatures for injection, transmission line, Quad, and ion source were 240 °C, 240 °C, 230 °C, and 150 °C, respectively. In the electron impact mode, the energy was −70 eV. After a solvent delay of 7 min, mass spectrometry data in the *m*/*z* range of 33–400 were obtained in the scanning/SIM mode.

### 4.9. DEGs and RNA Sequencing Mapping in BMECs of C4BPA Gene KO or WT Cell Line

The evaluation of expression disparities between WT-BMECs and KO-BMECs was conducted using the DESeq2 R package, renowned for its statistical algorithms that discern differential expression in numerical gene expression data through a negative distribution based model. The ensuing *p* values were computed employing the Benjamini–Hochberg method to regulate the error detection rate. DESeq2 identified significant differential expression (*p* < 0.05) in the target gene, with appropriate corrections applied. Interactive reading mapping, alternatively known as to-terminal reading, represents a technique for incremental mapping of readings on both sides. It is imperative to preprocess raw data accurately before initiating the mapping process, involving the removal of adapter sequences. The readings are deemed valid when the fuzzy base count exceeds 5% and is proportionate to a lower mass. A criterion of 20% or more bases below 20 is applied, and this figure is subsequently compared to the target genome.

### 4.10. GO and KEGG Analysis of DEGs

A cluster-profiler R package (3.4.3 Version) of differentially expressed genes (DEGs) (*p* < 0.05) was used to conduct an intensive GO analysis and the correction of the gene length bias. GO was significantly enriched by DEGs. Kyoto Encyclopedia of Genes and Genome Biology at a molecular level (KEGG) message system was used, with advanced features and utilization of database resources (https://www.genome.jp/kegg/pathway.html) (accessed on 10 September 2022) genome. DEGs in KEGG path were used to test for the accumulation of statistical cluster through a profiler R package. There was a significant accumulation of DEGs (*p* < 0.05) in the KEGG pathway.

### 4.11. Western Blotting

The protein expressions of HMGCS1, CPT1A, SCD1, and FADS1 in cells were determined by Western blotting. Briefly, cells were washed with phosphate buffered saline (PBS) twice and lysed in RIPA buffer containing phosphatase and protease inhibitors. Then, the samples were centrifuged at 12,000 rpm for 30 min at 4 °C. Proteins were separated after denaturation by SDS-PAGE and then transferred to polyvinylidene difluoride (PVDF) membranes (Millipore, Billerica, MA, USA). Then, the PVDF membranes were blocked with 5% skim milk and incubated overnight at 4 °C with the following primary antibodies: anti-HMGCS1 (1:2000, bs-5069R, Bioss), anti-CPT1A (1:1000, bs-23779R, Bioss), anti-SCD1 (1:1000, bs-3787R, Bioss), and anti-FADS1 (1:2000, bs-5060R, Bioss). The PVDF membranes were washed with TBST and incubated with secondary antibodies. Finally, protein bands were detected by using BeyoECL Plus (Beyotime Biotechnology Company in Shanghai, China). Grayscale scanning of the protein bands was quantified using Image J (NIH).

### 4.12. Statistical Analysis

Data analysis was conducted with GraphPad Prism 6.0 (GraphPad Software Inc., San Diego, CA, USA) and SPSS version 23.0 software (IBM Corporation, Armonk, NY, USA). A completely randomized Student’s *t* test was used for group comparisons. All data were presented as mean ± standard deviation (SD). The levels of statistical significance are expressed as (*) *p* < 0.05, (**) *p* < 0.01, and (***) *p* < 0.001. The assays were performed in triplicate.

## Figures and Tables

**Figure 1 ijms-25-02375-f001:**
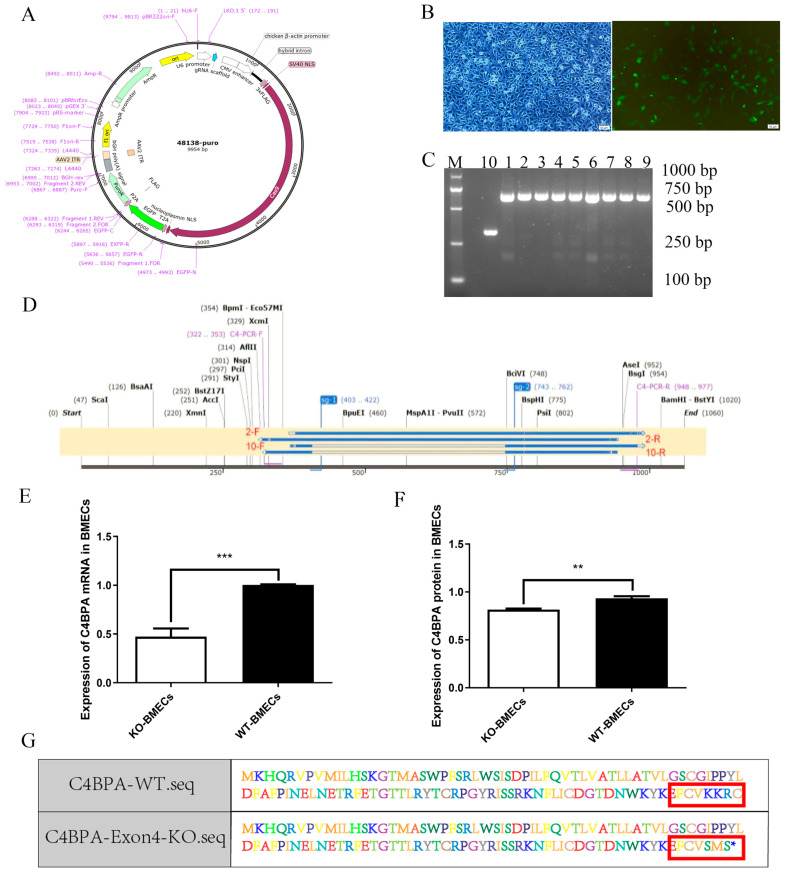
Construction of *C4BPA* gene knockout bovine mammary epithelial cells (BMECs) line. (**A**) Construction of the 48138-EGFP-puro vector; (**B**) transfection of bovine mammary epithelial cells with the sgRNA vector; (**C**) electrophoretogram illustrating the results of cell validation through PCR (lines 1–9 represent WT-BMECs, and number 10 corresponds to KO-BMECs); (**D**) DNA sequencing results for wild-type and knockout clones; (**E**) assessment of *C4BPA* mRNA levels in WT-BMECs and KO-BMECs using RT-qPCR analysis; (**F**) ELISA analysis revealed the *C4BPA* protein levels in WT-BMECs and KO-BMECs; (**G**) amino acid comparison results. Data are presented as means ± SD (*n* = 3). ** *p* < 0.01, and *** *p* < 0.001 compared with WT-BMECs.

**Figure 2 ijms-25-02375-f002:**
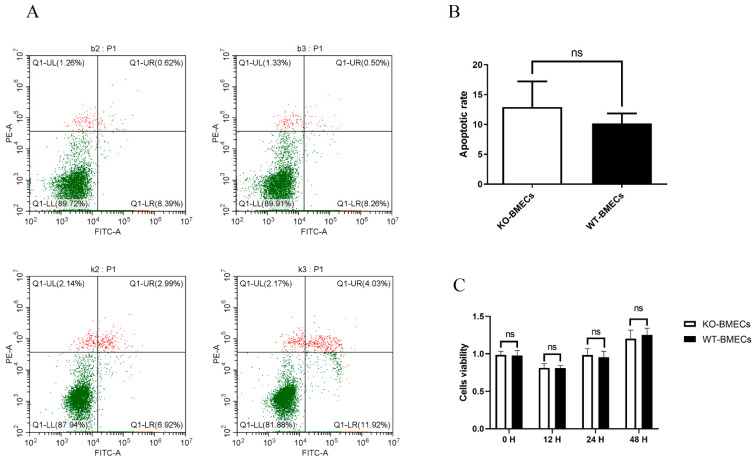
Effects of *C4BPA* knockout on cell proliferation and apoptosis. (**A**) The effect of *C4BPA* knockout on cell apoptosis, as detected by flow cytometry; The upper left quadrant (red) shows dead cells, the lower left quadrant (green) shows normal cells, the upper right quadrant (red) shows early apoptotic cells, and the lower right quadrant (green) shows late apoptotic cells. (**B**) bar graphs showing the apoptosis results; (**C**) changes in cell proliferation rate in different time gradients before and after *C4BPA* gene knockout. *p* = *ns* means that *p* > 0.05, there were no significant between WT-BMECs and KO-BMECs.

**Figure 3 ijms-25-02375-f003:**
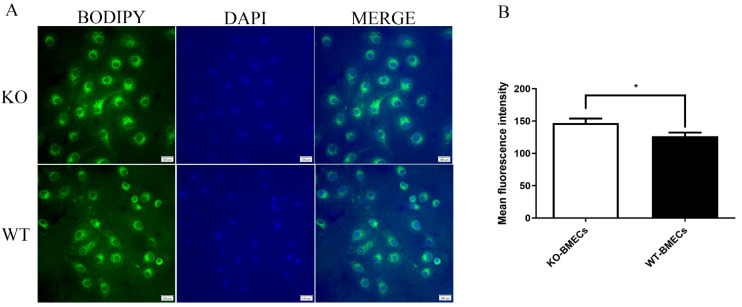
Effects of *C4BPA* gene knockout on lipid accumulation. (**A**) BODIPY staining was used to observe milk fat in BMECs. (**B**) Changes in lipid droplet expression before and after *C4BPA* gene knockout. Data presented as means ± SD (*n* = 3). * *p* < 0.05, compared with WT-BMECs. Scar bar = 20 μm.

**Figure 4 ijms-25-02375-f004:**
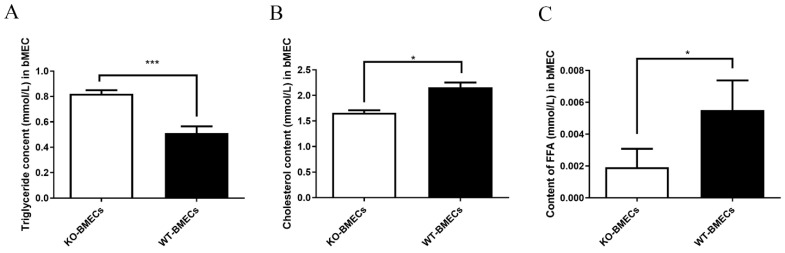
The effect of *C4BPA* knockout on TG, CHOL, and FFA contents in bovine mammary epithelial cells (BMECs). (**A**) Triacylglycerol content in WT-BMECs and KO-BMECs; (**B**) cholesterol content in WT-BMECs and KO-BMECs; (**C**) content of FFA in WT and KO-BMECs. Data presented as means ± SD (*n* = 3). * *p* < 0.05 and *** *p* < 0.001 compared with WT-BMECs.

**Figure 5 ijms-25-02375-f005:**
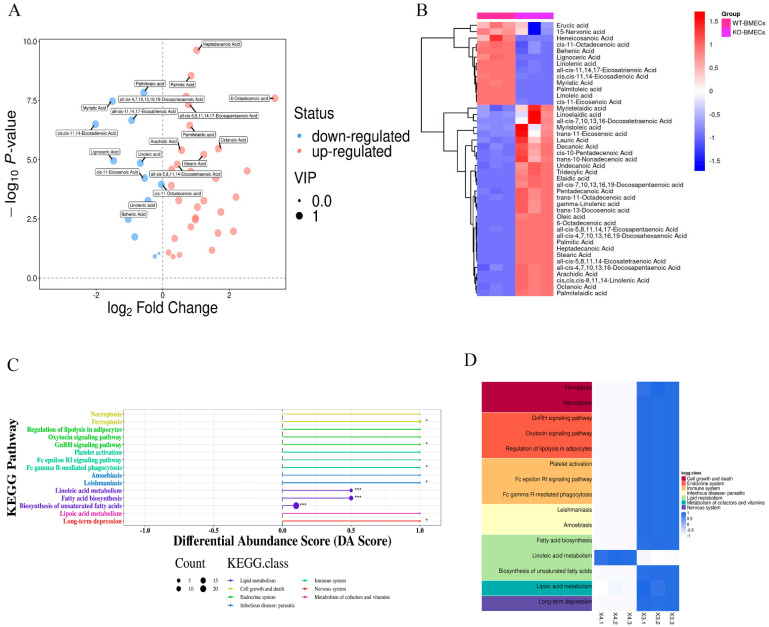
The role of *C4BPA* on fatty acid content in bovine mammary epithelial cells (BMECs). (**A**) Volcano plot for WT-BMECs and KO-BMECs; (**B**) heatmap of hierarchical clustering analysis for WT-BMECs and KO-BMECs; (**C**) differential abundance score for WT-BMECs and KO-BMECs; (**D**) KEGG heatmap for WT-BMECs and KO-BMECs. * *p* < 0.05 and *** *p* < 0.001 compared with WT−BMECs.

**Figure 6 ijms-25-02375-f006:**
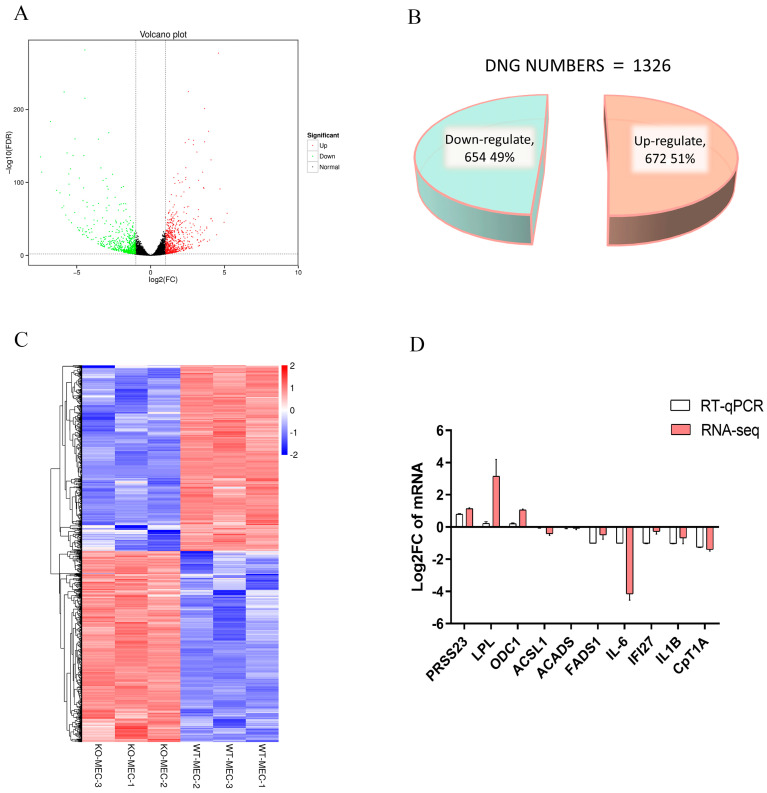
Identification of DEG in WT-BMECs and KO-BMECs by RNA-seq method. (**A**) Volcanic maps of WT-BMECs and KO-BMECs; (**B**) DEG counts downregulated and upregulated in WT-BMECs and KO-BMECs; (**C**) hierarchical clustering heat maps of WT-BMECs and KO-BMECs; (**D**) RT-qPCR and RNA-seq results of 10 genes to verify the RNA-seq analysis results of WT-BMECs and KO-BMECs.

**Figure 7 ijms-25-02375-f007:**
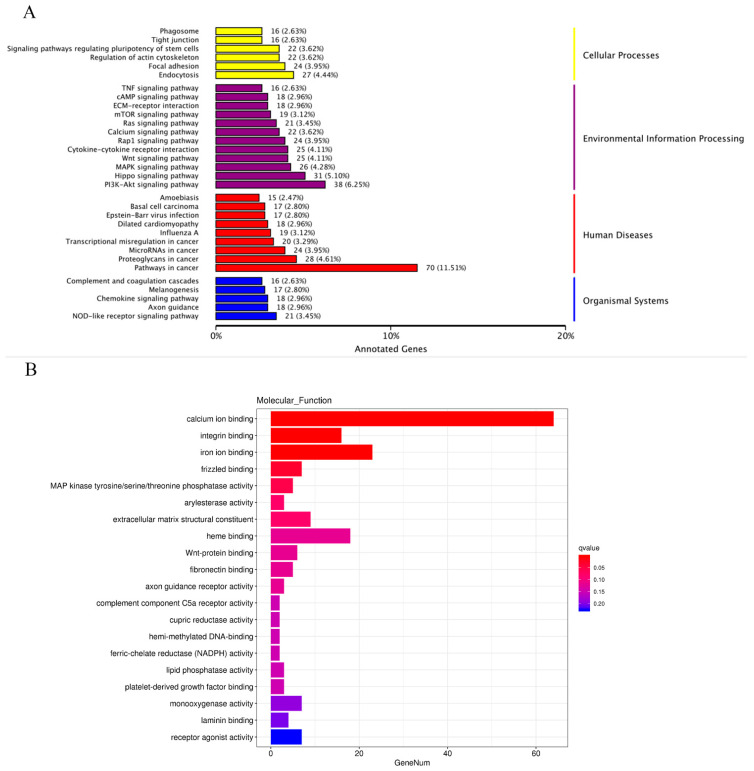
KEGGs and GO terms enrichment analysis. (**A**) The enrichment analysis of KEGG; (**B**) molecular functional enrichment analysis of GO pathway.

**Figure 8 ijms-25-02375-f008:**
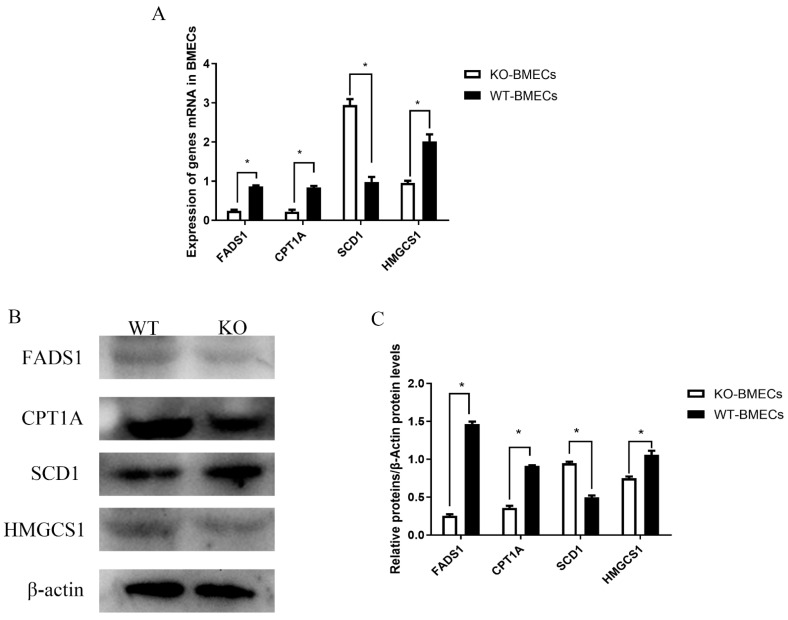
Relative expression of differentially expressed genes in KO-BMECs and WT-BMECs. (**A**) The mRNA expression of *HMGCS1*, *CPT1A*, *FADS1*, and *SCD1* genes; (**B**) the protein expression of HMGCS1, CPT1A, FADS1, and SCD1 genes; (**C**) the analysis results of gray intensity of HMGCS1, CPT1A, FADS1, and SCD1 genes. * *p* < 0.05 compared with WT-BMECs.

**Table 1 ijms-25-02375-t001:** SgRNA sequences of *C4BPA* gene.

SgRNA Strand	Target-Seq	Location	Gene ID
*C4BPA*-sgRNA1-F	CACCGAGAGAGCTGATGGATCATTG	Intron 3	281651
*C4BPA*-sgRNA1-R	AAACCAATGATCCATCAGCTCTCTC	Intron 3	281651
*C4BPA*-sgRNA2-F	CACCGAGTGTATCCCTGTTTAAAGA	Intron 5	281651
*C4BPA*-sgRNA2-R	AAACTCTTTAAACAGGGATACACTC	Intron 5	281651

## Data Availability

The raw data used for this study have been deposited in NCBI (https://www.ncbi.nlm.nih.gov/) and MetaboLights (https://www.ebi.ac.uk/metabolights/) with accession numbers PRJNA1053485 (RNA-Seq) and MTBLS9245 (GC-MS). All data analyzed can be found on the corresponding website.

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
