# Peer review of "The Complement Component 4 Binding Protein α Gene: A Versatile Immune Gene That Influences Lipid Metabolism in Bovine Mammary Epithelial Cell Lines"

_ijms, 2024, doi:10.3390/ijms25042375_

Round 1

Reviewer 1 Report

Comments and Suggestions for Authors
  1. What are the specific sgRNAs used to target C4BPA in BMECs, and how were they designed?

  2. Can you provide more details on the transfection process of the recombinant plasmid into BMECs? Were any transfection efficiency assessments conducted?

  3. How was the agarose gel electrophoresis performed to confirm the successful disruption of C4BPA in KO-BMECs? Were any controls included in the gel?

  4. What specific mutations or alterations were identified through genomic DNA sequencing, which confirmed the disruption of C4BPA target sequences in BMECs?

  5. Can you elaborate on the methods used for mRNA and protein expression analysis in both WT-BMECs and KO-BMECs? Were multiple replicates performed?

  6. What downstream effects or consequences were observed due to the significant reduction in C4BPA mRNA and protein expression in KO-BMECs? Were any functional assays or phenotypic assessments conducted?

  7. Could you provide more details about the amino acid sequence differences between the protein sequences of WT-BMECs and KO-BMECs? Were any potential structural or functional implications discussed?

  8. Did you observe any unexpected or off-target effects resulting from the CRISPR/Cas9-mediated disruption of C4BPA in BMECs?

  9. I recommend referencing the following paper in the introduction section. For further details, please consult the provided reference below.

  10. Multiple omics analysis reveals the regulation of SIRT5 on mitochondrial function and lipid metabolism during the differentiation of bovine preadipocytes. Genomics116(1), p.110773.
  11. FOXO1 regulates the formation of bovine fat by targeting CD36 and STEAP4. International Journal of Biological Macromolecules248, p.126025.
  12. Interaction of C/EBPβ with SMAD2 and SMAD4 genes induces the formation of lipid droplets in bovine myoblasts. Functional & Integrative Genomics23(2), 191.
  13. Association of variants and expression levels of MYOD1 gene with carcass and muscle characteristic traits in domestic pigeons. Animal Biotechnology, 1-11.
  1. What are the potential applications or implications of these findings in the context of C4BPA and its role in BMECs or related biological processes?

  2. Are there any plans for further experiments or studies to investigate the specific mechanisms underlying the observed differences between WT-BMECs and KO-BMECs?

Reviewer 2 Report

Comments and Suggestions for Authors

The article by Chen et al described the role of C4BPA in influencing lipid metabolism in BMECs. CRISPR/Cas9-based C4BPA gene knockout BMECs demonstrated an increase in triglycerides and a decrease in cholesterol and free fatty acids. KEGG analysis showed the enrichment of fatty acids implicated in the biosynthesis of unsaturated fatty acids, linoleic acid metabolism, fatty acid biosynthesis, and regulation of lipolysis. RNA-seq analyses identified the PI3k/Akt signaling pathway as the most significantly affected pathway due to C4BPA knockout. Several genes associated with lipid metabolism were also downregulated following C4BPA knockdown. Overall, this study built upon their prior research and delineated a metabolic role of C4BPA, which is interesting. However, several concerns need to be addressed.

1.       Why did the authors use sgRNA targeting the exon 4? Is the targeted region important for the proper function of C4BPA? In other words, did the authors check whether the canonical immune functions of C4BPA were affected in the new C4BPA variant following CRIPS/Cas9 mediated deletion?

2.       Did the authors use another sgRNA targeting a different region to confirm the observed metabolic effects were indeed on-target instead of being some off-target effects?  

3.       The change in lipid accumulation following C4BPA knockdown in Fig. 3 is not convincing. The DAPI background intensities are not similar between WT and KO cells, suggesting that the images may not be captured in the same instrumental settings.

4.       The authors mentioned the quality of milk products as the rationale for studying the C4BPA gene in BMECs but did not include any discussion of how the outcome of C4BPA depletion such as a change in triglycerides, cholesterol, or free fatty acids would affect the quality of milk.

Minor points:

5.        Fig 2C showed data for up to 48 hours, but the results section mentioned up to 24 hours (Page 4, Line 97).

6.       Page 8 Line 167-169 should be Figure 7A and 7B respectively.

Round 2

Reviewer 2 Report

Comments and Suggestions for Authors

The authors have provided satisfactory responses to my concerns raised in the prior round of review. I do not have any further comments.